

# Effects of temperature and salinity stress on DNA methylation in a highly invasive marine invertebrate, the colonial ascidian *Didemnum vexillum*

Nicola A. Hawes[1,2], Louis A. Tremblay[2,3], Xavier Pochon[1,2], Brendon Dunphy[1,3], Andrew E. Fidler[1] and Kirsty F. Smith[2]

[1] Institute of Marine Science, University of Auckland, Auckland, New Zealand
[2] Cawthron Institute, Nelson, New Zealand
[3] School of Biological Sciences, University of Auckland, Auckland, New Zealand

Corresponding author
Nicola A. Hawes,
Nicola.hawes@cawthron.org.nz

## ABSTRACT

Environmentally induced epigenetic changes may contribute to phenotypic plasticity, increase adaptive potential in changing environments, and play a key role in the establishment and spread of invasive species in new habitats. In this study, we used methylation-sensitive amplified polymorphism (MSAP) to assess environmentally induced DNA methylation changes in a globally invasive clonal ascidian, *Didemnum vexillum*. We tested the effect of increasing temperature (19, 25 and 27 °C) and decreasing salinity (34, 32, 30, 28 and 26 practical salinity units (PSU)) on global DNA methylation, growth and survival rates. Exposure to 27 °C resulted in significant changes in DNA methylation over time. Growth also decreased in colonies exposed to high temperatures, suggesting they were under thermal stress. In contrast, no differences in growth nor DNA methylation patterns were observed in colonies exposed to a decreasing salinity gradient, potentially due to prior adaptation. The results of this study show that environmental stress can induce significant global DNA methylation changes in an invasive marine invertebrate on very rapid timescales, and that this response varies depending on the type, magnitude, and duration of the stressor. Changes in genomic DNA methylation and the rate of growth may act to 'buy survival time' under stressful conditions, expanding the distribution limits of this globally invasive species.

## INTRODUCTION

Species invasions, climate change, habitat fragmentation and environmental degradation are altering ecosystems and threatening biodiversity (*Leadley, 2010*). A key question in evolutionary biology is whether species will be able to adapt in response to these human-driven environmental changes (*Visser, 2008*). Biological invasions can provide a unique model to investigate adaptation and evolution within short timescales, as introduced species must rapidly adapt to new habitats (*Allendorf & Lundquist, 2003*; *Sakai et al., 2001*). It has been suggested that epigenetic mechanisms could play a critical role in

environmental adaptation, and may be particularly important for the success of invasive species (*Estoup et al., 2016*; *Pérez et al., 2006*; *Prentis et al., 2008*). Recently introduced populations frequently have reduced genetic diversity (e.g., genetic bottlenecks and founder effects) (*Dlugosch & Parker, 2008*), which is thought to constrain the colonisation potential of a species (e.g., *Crawford & Whitney, 2010*). Despite this, invasive species can still be highly successful in their new environments, and often outcompete locally adapted native species (*Allendorf & Lundquist, 2003*). By increasing both phenotypic plasticity and heritable variation, epigenetic changes might allow invasive species to quickly respond to environmental challenges. However, the role that epigenetic mechanisms play during the process of invasion is only beginning to be understood (*Hawes et al., 2018*) and, for many species, the effect of environmental stressors on DNA methylation is unknown.

Epigenetic modifications have been shown to respond to environmental cues and, in some cases, be associated with significant phenotypic change (*Dias & Ressler, 2014*; *Kucharski et al., 2008*; *Waterland & Jirtle, 2003*). Epigenetic mechanisms are diverse and interactive (e.g., DNA methylation, histone modifications, small RNAs), but all alter gene expression without the requirement for changes in the underlying DNA nucleotide sequences (*Bossdorf, Richards & Pigliucci, 2008*). Currently the most studied epigenetic mechanism is the methylation of cytosine nucleotides to form 5-methylcytosine (DNA methylation). DNA methylation is common in eukaryotes, and there are a range of methods for its detection and quantification (*Plongthongkum, Diep & Zhang, 2014*). One such method, methylation-sensitive amplified polymorphism (MSAP), allows for cost-effective screening of variation in global DNA methylation, without the requirement for a reference genome (*Reyna-Lopez, Simpson & Ruiz-Herrera, 1997*). The MSAP technique enables epigenetic research in non-model organisms and can provide a first look at DNA methylation-environment interactions, which may underlie adaptive plasticity. Interest in the ecological relevance of DNA methylation in non-model organisms is growing, and marine invertebrates have been identified as an emerging taxonomic group for studies of ecological epigenetics, particularly in the context of environmental change (*Hofmann, 2017*). Despite this, few studies have investigated environmentally induced epigenetic changes in marine invertebrates (but see *Marsh, Hoadley & Warner, 2016*; *Marsh & Pasqualone, 2014*; *Putnam, Davidson & Gates, 2016*), and the role of epigenetic mechanisms in the success of marine invertebrate invaders is only beginning to be explored (*Ardura et al., 2017*; *Huang et al., 2017*; *Pu & Zhan, 2017*).

Of the marine invertebrates, colonial ascidians stand out as model species to study both environmentally induced DNA methylation changes (*Hawes et al., 2018*) and invasion success (*Zhan et al., 2015*). Colonial ascidians (phylum Chordata) are common marine invaders worldwide, particularly in habitats perturbed by human activities (e.g., marinas, ports, aquaculture structures) (*Lambert, 2001*). Due to the considerable ecological and economic damage caused by ascidian invasions, they have become a prominent study species in the field of invasion biology (*Zhan et al., 2015*). Ascidians can thrive in a variety of environmental conditions, and display unique biological characteristics, including a broad tolerance to common environmental stressors such as temperature and salinity (*Rocha, Castellano & Freire, 2017*). Additionally, the germ-cell lineages of colonial

ascidians originate from somatic-cell lineages, contrasting with the germ-cell lineage sequestration found in vertebrate Chordates (*Rosner et al., 2009*). Having no true germ-cell lineage sequestration increases the likelihood that stress-induced epigenetic modifications induced in somatic cells can be passed on to gametes (*Verhoeven & Preite, 2014*). Finally, asexual reproduction (by budding) leads to colonies of genetically identical individual animals (termed zooids) that all share the same DNA nucleotide sequences (genotype). Clonal reproduction allows for genetically identical replicates across environment stress treatments, and repeated sampling of the same individual at multiple time-points. This reduces the confounding effects of genetic variation that frequently complicate epigenetic studies of non-clonal organisms (*Douhovnikoff & Dodd, 2015*; *Verhoeven & Preite, 2014*).

Invasive populations of the colonial ascidian, *Didemnum vexillum* Kott, 2002 have extremely low levels of genetic diversity compared to populations within its native range (*Stefaniak et al., 2012*). Despite this, *D. vexillum* is extremely successful where it has invaded (*Beveridge et al., 2011*; *Cohen et al., 2011*; *Griffith et al., 2009*; *Hitchin, 2012*; *Lambert, 2009*; *Tagliapietra et al., 2012*), often forming large colonies that smother other marine invertebrates, including commercial aquaculture species (*Fletcher, Forrest & Bell, 2013b*). We used MSAP to determine whether (a) DNA methylation is present in the genome of *D. vexillum*, and (b) if genome-wide DNA methylation patterns in *D. vexillum* change in response to two prominent types of environmental stress: temperature and salinity. Temperature and salinity are often reported as the most important environmental determinants controlling the distribution of marine species, and these two parameters have been used repeatedly when studying the tolerance of ascidians to environmental stress (*Dybern, 1967*; *Gröner et al., 2011*; *Renborg, Johannesson & Havenhand, 2014*; *Serafini et al., 2011*; *Zerebecki & Sorte, 2011*). Furthermore, extreme climatic events such as precipitation events and heatwaves are expected to increase in frequency in the near future (*IPCC, 2014*), making it increasingly important to evaluate the response and resilience of marine invertebrates to thermal and osmotic stress.

## MATERIALS AND METHODS

### Sample collection and establishment of experimental colonies

Colonies of *D. vexillum* were collected from the Nelson Marina (South Island, New Zealand; 41°15′38″S, 173°16′54″E) in April 2016. At the time of collection, the water temperature was 19 °C and the salinity was 33 PSU. Colonies were gently removed from wharf pilings and immediately placed in labelled 2 L plastic containers filled with ambient seawater for transport to the Cawthron Institute (less than 5 min commute). Attempts were made to remove whole colonies that were free from debris, but as the colonies were mostly small and growing flat over piles covered in other fouling, colonies often broke apart during removal. We treated these fragments as one colony for the following experiments but it is possible that it was two or more colonies growing in close proximity. After arrival at the laboratory, the colonies were gently cleaned with seawater to remove mud, silt and other organisms, and approximately equal sized fragments (c. 1.5 cm × 1.5 cm) were cut with a razor blade. Colony fragments were then placed on glass slides and gently wrapped with cotton thread

to encourage attachment (*Rinkevich & Fidler, 2014*). Glass slides were inserted into slide holders and placed in pre-conditioned 40 L glass aquaria in ambient, control conditions (19 °C, 34 PSU) for one week to allow attachment to occur. During this acclimation period, colonies were fed $1.6 \times 10^8$ cells $L^{-1}$ of cultured algae (*Isochrysis galbana*) every second day. Following attachment, the cotton was removed and the colony fragments were randomly allocated (Temperature: $n = 9$, Salinity: $n = 15$) to pre-conditioned 5 L treatment tanks ($n =$ three tanks per treatment, one colony fragment per treatment tank). All treatment tanks were maintained at 19 °C and 34 PSU for a further two weeks acclimation time prior to beginning the experiments.

## Experimental system

Seawater for the experiments was collected from Tasman Bay, Nelson (41°11′29.2″S 173°21′01.9″E), passed through three filters (pore size 50, 5, and 0.35 µm) and was ultraviolet light treated. Each tank was filled with freshly collected seawater at the start of the experiment, and 5 L water exchanges were done daily throughout the duration of the experiment with pre-heated or reduced salinity seawater. Each day following water exchange, colonies were fed a diet of $1.6 \times 10^8$ cells $L^{-1}$ of *I. galbana*. To prevent stratification, mixing in experimental tanks was ensured by gentle aeration using air stones. Experimental tanks were exposed to a 14:10 h light:dark cycle to mimic summer conditions. Water temperatures were maintained using thermostatically regulated aquarium heaters (EHEIM JAIGER 100W, Deizisau, Germany) and salinity treatments were achieved and maintained by the addition of reverse osmosis (RO) water to seawater. Water temperature and salinity were measured twice daily using hand-held probes to ensure stable treatment conditions were maintained ±0.5 °C or 0.5 PSU (YSI Professional Plus, YSI Incorporated, Yellow Springs, Ohio, USA). After two weeks of acclimation, tissue samples were collected from all colonies for MSAP analyses (Time 0; T0). Temperatures were then increased by 1 °C per day for temperature treatments, and salinity was reduced by 1 PSU per day for the salinity treatments until, after eight days, all treatment conditions were reached. Treatments were as follows: Temperature = 19 (control), 25, and 27 °C and Salinity = 34 (control), 32, 30, 28, and 26 PSU. Temperature and salinity experiments were run in parallel. Tissue samples were again taken at day 8 for MSAP analyses (Time 1; T1). Colony fragments were then maintained in experimental treatments for a further three days (day 11), at which time final tissue samples were taken for MSAP analyses (Time 2; T2).

## Sampling protocol

Prior to tissue collection, colonies were not fed for 16 h to minimise contamination by feed microalgae. To collect tissue for MSAP analyses, small (c. 5 mm × 5 mm) samples were taken from each colony using a sterile razorblade. Tissue samples were preserved in 95% ethanol, which was refreshed once before storage at −20 °C until processing. Pre- and post-tissue sampling, photos were taken for growth rate calculations. Colony growth rates (quantified by changes in colony surface-area over time) were measured using Image J 1.48v software (*Schneider, Rasband & Eliceiri, 2012*). The survival of colonies was also monitored at the time of sampling by assessments of colony health, including zooid

**Table 1**  Adapter and primer sequences used for MSAP protocol.

|  | Sequence 5′–3′ |
|---|---|
| **Adapters** | |
| *Eco*RI-adapter F | 5′-CTC GTA GAC TGC GTA CC-3′ |
| *Eco*RI-adapter R | 5′-AAT TGG TAC GCA GTC TAC-3′ |
| *Hpa*II and *Msp*I-Adapter F | 5′-GAC GAT GAG TCT AGA A-3′ |
| *Hpa*II and *Msp*I-Adapter R | 5′-CGT TCT AGA CTC ATC-3′ |
| **Pre-selective primers** | |
| *Eco*RI-A | 5′-GAC TGC GTA CCA ATT CA-3′ |
| *Hpa*II and *Msp*I –T | 5′-GAT GAG TCT AGA ACG GT-3′ |
| **Selective primers** | |
| *Eco*RI + AAG | 5′-GAC TGC GTA CCA ATT CAA G-3′ |
| *Eco*RI + ACT | 5′-GAC TGC GTA CCA ATT CAC T-3′ |
| *Hpa*II *Msp*I + TAC | 5′-6-FAM-GAT GAG TCT AGA ACG GTA C-3′ |
| *Hpa*II *Msp*I + TCC | 5′-6-FAM-GAT GAG TCT AGA ACG GTC C-3′ |

integrity, colour and texture of the colony, build-up of detritus and dead tissue. Throughout the experiment, all colonies were transported, sampled and photographed while submerged in trays of temperature and salinity adjusted seawater to minimise handling stress.

## MSAP analysis

To assess whole genome DNA methylation patterns, DNA was extracted using G-spin Total DNA extraction kits (animal tissue protocol; Intron, Gyeonggi-do, South Korea). Following DNA extraction, in parallel reactions (25 µL final reaction volume), 500 ng of DNA was digested with 10 U of each restriction enzyme (MspI and EcoRI or HpaII and EcoRI; New England BioLabs, Ipswich, MA, USA) and 10X CutSmart Buffer (New England BioLabs), and incubated at 37 °C for 2 h followed by 80 °C for 20 min to inactivate the enzymes. The digested DNA were ligated in a final volume of 20 µL containing 1 U of T4 DNA ligase (New England BioLabs), 10X ligase buffer, 250 nM of EcoRI adapter, and 2.5 µM of MspI or HpaII adaptor for 3 h at 37 °C. Table 1 presents a list of all adapter and primer sequences used for the MSAP protocol. Pre-selective polymerase chain reaction (PCR) was performed in a total volume of 20 µL using 8 µL of ligated DNA, MyTaq^TM 2X PCR master mix (Bioline, Taunton, MA, USA), and 500 nM of each pre-selective primer. Thermocycling conditions were 20 cycles of: 94 °C for 30 s, 56 °C for 60 s, and 72 °C for 60 s. Selective PCR was performed using four EcoRI and MspI/HpaII primer combinations in a final volume of 20 µL using 1 µL of pre-selective PCR product, MyTaq^TM 2X PCR master mix (Bioline, Taunton, MA, USA), and 500 nM of each combination of forward and reverse selective primers. Thermocycling conditions were one cycle of: 94 °C for 2 min; 10 cycles of 94 °C for 30 s, 65 °C for 30 s (decreasing 1 °C per cycle), and 72 °C for 60 s; and 30 cycles of 94 °C for 30 s, 56 °C for 30 s, and 72 °C for 60 s; and a hold cycle of 72 °C for 30 min. The resulting selective PCR product was diluted 1:5 with sterile distilled water and analysed using an ABI 3130 capillary sequencer (Applied Biosystems, Foster City, CA, USA) with internal size standards (GS600LIZ) by an external contractor (Genetic Analysis Services, University of Otago, Dunedin, New Zealand).

**Table 2  CCGG sites where methylation sensitive restriction enzymes (HpaII and MspI) cleave (Yes) or do not cleave (No) to generate methylation dependent fragment patterns.** Both MspI and HpaII recognise CCGG sites and cleave unmethylated CCGG sites (1/1), but MspI cannot cleave when the outer cytosine is fully or hemimethylated (m), and HpaII cannot cleave when the inner or outer cytosine is methylated on both strands. Cleaving by both enzymes is blocked when both cytosines are methylated. (0/0), which is considered uninformative as it could be due to either hypermethylation or fragment absence. From this the methylation state of restriction sites can be scored (e.g., methylated (1/0 or 0/1) unmethylated (1/1) and uninformative (0/0)).

| | Restriction sites | | HpaII | MspI | Fragment classification |
|---|---|---|---|---|---|
| Type I | 5′-CCGG GGCC-5′ | | Yes | Yes | Unmethylated 1/1 |
| Type II | 5′-C$^m$CGG GG$^m$CC-5′ | | No | Yes | Internal cytosine methylation 0/1 |
| Type III | 5′-$^m$CCGG GGCC-5′ | 5′-$^m$C$^m$CGG GGCC-5′ | Yes | No | Hemimethylated 1/0 |
| Type IV | 5′-$^m$C$^m$CGG GG$^m$C$^m$C-5′ | 5′-$^m$CCGG GGC$^m$C-5′ | No | No | Uninformative 0/0 |

## Data analysis

PEAKSCANNER software (Applied Biosystems, Foster City, CA, USA) was used to assign the MSAP fragments peak height and size. To determine the parameters for subsequent analysis, the MSAP procedure described above was first repeated three times for one individual sample. The following settings were associated with the lowest error rate between replicates and were applied to the *msap* analysis (*Pérez-Figueroa, 2013*) described below. Error rate per primer, 0.07; analysis range, 50–500 base pairs (bp); minimum peak height, 1,200 relative fluorescence units. Peak presence/absence data corresponding to HpaII and MspI fragments was then converted to a binary matrix (presence = 1, absence = 0), so that the methylation state of each restriction site could be identified. MSAP profiles were assessed using the R package msap v. 1.1.8 (*Pérez-Figueroa, 2013*). The *msap* package determines whether individual fragments (loci) are methylated (MSL) by analysing the contents of the binary matrix, and comparing differences representing the differential sensitivities of HpaII and MspI to cytosine methylation (Table 2). From this, DNA methylation profiles of control and experimental samples were assessed by means of principal coordinate analyses (PCoA) followed by analyses of molecular variance (AMOVA) (*Excoffier, Smouse & Quattro, 1992*). Colony growth rates (mean growth per day, mm ±1 s.e.) were assessed using standard ANOVA and post-hoc pairwise comparisons were made using Tukey's honest significance difference (HSD) test.

## RESULTS

### Temperature stress experiments
#### Growth rates
All colonies survived elevated temperature exposure, and colony growth rates (Fig. 1) were significantly different between treatments (ANOVA, $F(2,6) = 7.82$, $p = 0.0213$; Table S1). Post hoc comparisons using Tukey's HSD indicated that growth per day was significantly reduced when colonies were exposed to 27 °C compared to colonies grown at 19 °C

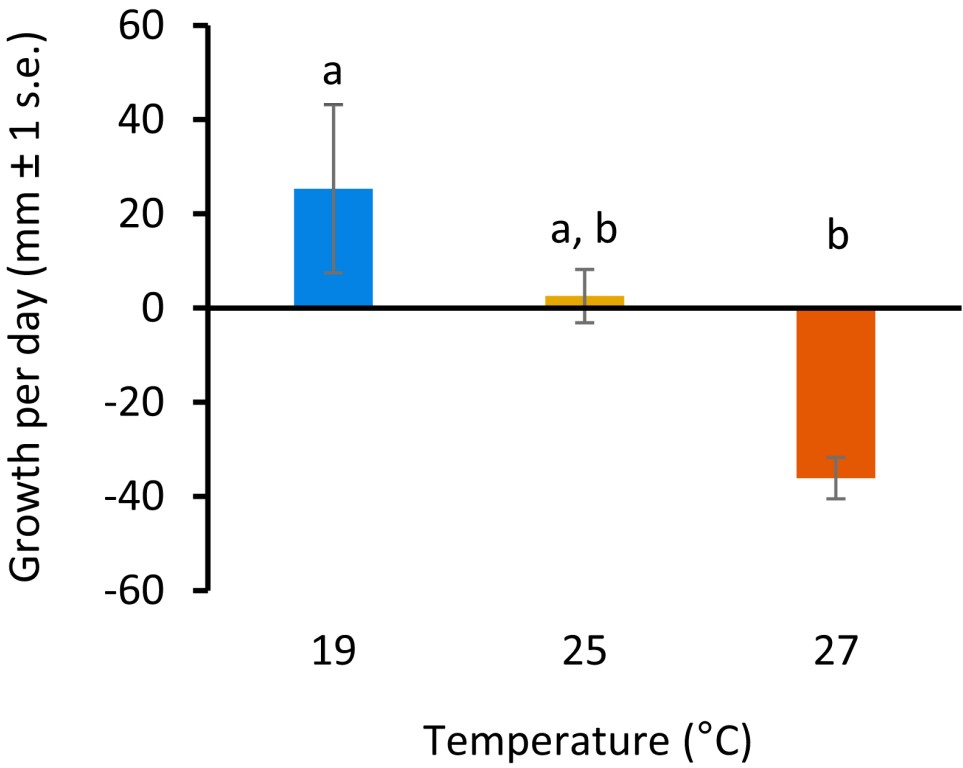

**Figure 1** **Colony growth rates with increasing temperature.** Colony growth rates (mean growth per day, mm ± 1 s.e.) at 19, 25 and 27 °C. Significant differences between treatments are denoted by different letters ($p < 0.05$, Tukey's test HSD).

(Diff $= -61.43$, $p = 0.01$; Table S2, Fig. 1). Growth was not significantly different between colonies exposed to 25 °C and 27 °C or 19 °C and 25 °C.

### Whole genome DNA methylation patterns (MSAP)

Using four primer combinations, 1,157 fragments (loci) were produced and analysed. At sampling Time 0 (T0), prior to temperature treatment exposure, 586 of the 1,157 loci were MSL, of which 178 were polymorphic (30%). There were no significant differences in DNA methylation patterns (Fig. 2A) (MSL, $\phi_{ST} = -0.04981$, $p = 0.7093$, AMOVA; Table S3) between treatment groups. Following eight days of gradual temperature increase at Time 1 (T1), of the 1,157 loci analysed, 613 were MSL, of which 172 were polymorphic (28%). There were still no significant differences in DNA methylated loci (MSL) (Fig. 2B) ($\phi_{ST} = 0.05606$, $p = 0.2176$; Table S3) between the three temperature treatments. However, after three days of exposure to elevated temperature, of 1,157 loci, 600 were MSL, of which 201 were polymorphic (34%), and statistically significant differences in DNA methylation were evident between treatment groups (Fig. 2C) (MSL, $\phi_{ST} = 0.1585$, $p = 0.0215$; Table S3). There were no significant global methylation changes between sampling time points in colonies held at 19 °C (Fig. 2D) (MSL, $\phi_{ST} = -0.0212$, $p = 0.6019$; Table S5) or 25 °C (Fig. 2E) (MSL, $\phi_{ST} = -0.01373$, $p = 0.5637$; Table S4). In contrast, there were significant DNA methylation changes following exposure to 27 °C (Fig. 2F) (MSL, $\phi_{ST} = 0.1727$, $p = 0.0223$;

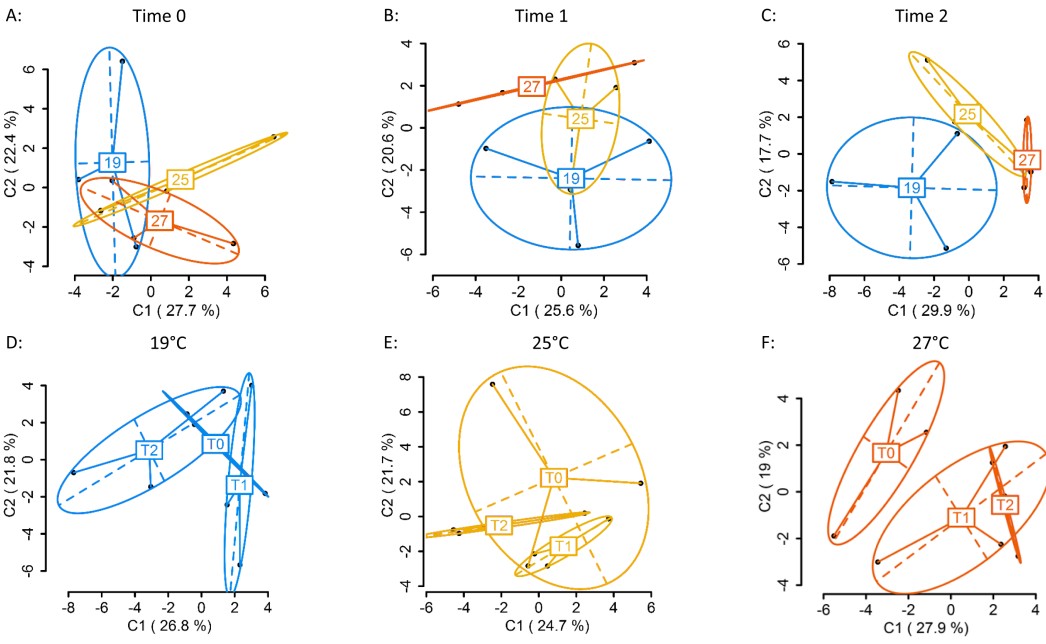

**Figure 2** **Principal Coordinate Analysis (PCoA) of methylation (MSL) differences between colonies exposed 19, 25 and 27 °C.** (A) between colonies at Time 0 (T0), prior to elevated temperature exposure (baseline methylation); (B) between colonies at Time 1 (T1) following a gradual temperature increase of 1 °C per day until all treatment temperatures were reached: 19 °C (control) ($n = 3$), 25 °C ($n = 3$) and 27 °C ($n = 3$); (C) at Time 2 (T2)* after three days of elevated temperature exposure; (D) between sampling time points (T0, T1, T2) in colonies held at 19 °C; (E) 25 °C; (F) 27 °C*. The first two coordinates (C1 and C2) are shown with the percentage of variance explained by them. Points in each group cloud represent individuals from different groups. Temperature labels show the centroid for the points cloud in each group. Ellipses represent average dispersion of those points around their centre (*Pérez-Figueroa, 2013*). AMOVA tests for significant differences in methylation (MSL) are shown in Tables S3 and S4. * represents significant difference ($p < 0.05$).

Table S4). Variation between individuals was also reduced in the 27 °C treatment group, with DNA methylation patterns becoming more similar, as visualised by the reduced spread of samples around the centroid in the PCoA for each temperature (Fig. 2C).

## Salinity stress experiments
### Growth rates
All colonies survived decreased salinity exposure, and colony growth rates were not significantly different between treatment groups (Fig. 3) (ANOVA, $F(4, 10) = 2.066$, $p = 0.161$; Table S5).

### Whole genome DNA methylation patterns (MSAP)
Using four primer combinations, 1,050 loci were produced and analysed. At sampling Time 0 (T0), prior to differential salinity exposure, of these, 626 were methylation sensitive loci (MSL), of which 169 were polymorphic (27%). There were no significant differences in DNA methylation between treatment groups (Fig. 4A) (MSL, $\phi_{ST} = -0.05023$, $p = 0.686$; Table S6). Following the gradual salinity decrease, at Time 1 (T1), of 1,050 loci 654 were

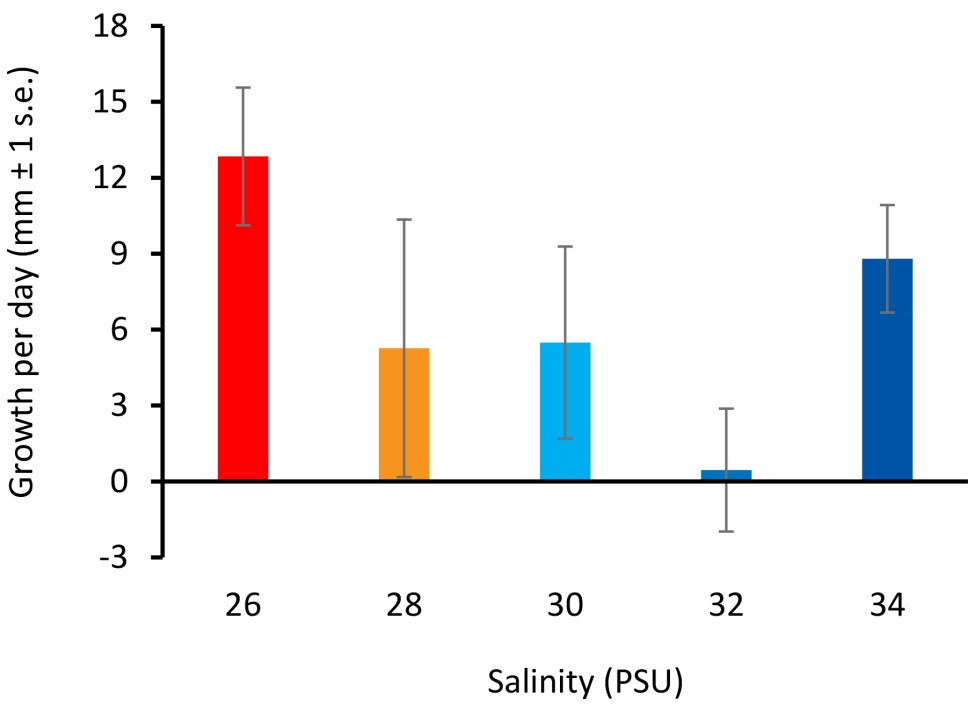

**Figure 3** **Colony growth rates with decreasing salinity.** Colony growth rates (mean growth per day, mm ± 1 s.e.) at 26, 28, 30, 32, and 34 PSU.

MSL, of which 173 were polymorphic (26%). There were still no significant differences in DNA methylation (Fig. 4B) (MSL, $\phi_{ST} = 0.02674$, $p = 0.3518$; Table S6), between salinity treatments. After three days of exposure to elevated salinity, of 1,050 loci, 682 were MSL, of which 95 were polymorphic (14%). Non-significant differences in DNA methylation remained (Fig. 4C) (MSL, $\phi_{ST} = -0.06389$, $p = 0.8328$). There were no significant methylation differences (Table S7) in any of the salinity treatments over time (Figs. 4D–4H).

## DISCUSSION

Changes in DNA methylation may be one of the mechanisms by which invasive species can rapidly adapt to new environments. However, for many species, the responsiveness of DNA methylation to environmental challenges has not yet been tested. Our results indicate that environmental stressors can induce significant global DNA methylation changes in an invasive marine invertebrate on very rapid timescales, and that this response varies depending on the type, magnitude, and duration of the stressor. After three days of exposure to elevated temperature, significant changes in whole-genome patterns of DNA methylation had occurred in *D. vexillum* colonies held at 27 °C. In contrast, DNA methylation patterns in colonies exposed to 25 °C and 19 °C did not change significantly over time. It is yet to be tested if significant changes would be observed at 25 °C if the duration of exposure was extended, but our results provide the first indication of methylation divergence with

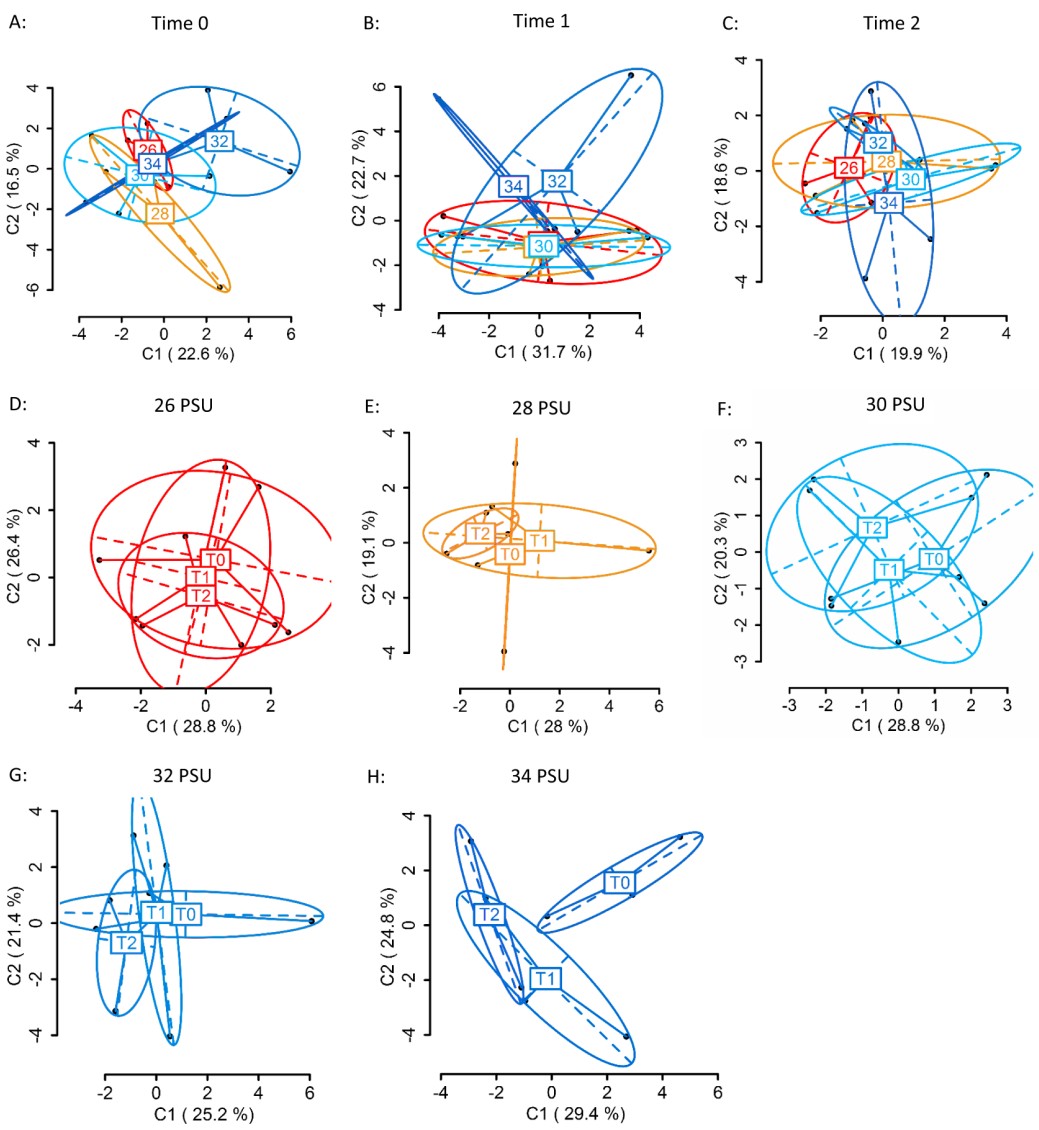

**Figure 4** **Principal Coordinate Analysis (PCoA) of methylation (MSL) differences between colonies exposed to 34, 32, 30, 28 and 26 PSU.** (A) between colonies at Time 0 (T0), prior to salinity treatment exposure (baseline methylation); (B) between colonies at Time 1 (T1) following a gradual salinity decrease of 1 PSU per day until all salinity treatments were reached: 34 PSU (control) ($n = 3$), 32 PSU ($n = 3$), 30 PSU ($n = 3$), 28 PSU ($n = 3$) and 26 PSU ($n = 3$); (C) at Time 2 (T2) after three days of decreased salinity exposure; (D) between sampling time points (T0, T1, T2) in colonies held at 26 PSU; (E) 28 PSU; (F) 30 PSU; (G) 32 PSU; (H) 34 PSU. The first two coordinates (C1 and C2) are shown with the percentage of variance explained by them. Points in each group cloud represent individuals from different groups. Labels show the centroid for the points cloud in each group. Ellipses represent average dispersion of those points around their centre (*Pérez-Figueroa, 2013*). AMOVA tests for significant differences in methylation (MSL) are shown in Tables S6 and S7. No significant differences between groups were found.

increasing temperature, and this effect may increase with time. In contrast, we did not find any significant DNA methylation changes in response to the salinity treatments used in this study.

*D. vexillum* is a subtidal species and can tolerate severe, short-term declines in salinity, but extended periods of low-salinity stress lead to mortality (*Gröner et al., 2011*) and ascidians are rarely found in salinities lower than 25 PSU (*Lambert, 2005*). Based on the above, we selected five salinity treatments: 34 PSU (within the upper range at collection site), 32 PSU, 30 PSU (within the middle range at the collection site), 28 PSU (within the lower range at the collection site) and 26 PSU (ascidians are rarely found below this level globally). However, at the colony collection site for this study (the Nelson marina), salinity frequently drops below 26 PSU (*Atalah, 2017*). The Nelson marina is located within 1.5 km of a river mouth, and salinity drops are likely associated with rain events (*Fletcher et al., 2013a*). The lack of response of genomic DNA methylation to changes in salinity suggests that *D. vexillum* in the Nelson marina may already be adapted to a lower salinity environment than was assessed in this experiment. This result is supported by the positive growth of colonies in all salinity treatments and a lack of negative health indicators, an indication that colonies were not experiencing significant stress. However, non-significant global methylation changes do not necessarily demonstrate that important DNA methylation changes are not occurring. Locus specific methylation differences have been associated with environmental differences in temperature and salinity in solitary ascidians (*Pu & Zhan, 2017*), and MSAP results can be difficult to interpret due to changes in many genes at once.

One other study has experimentally investigated the effect of environmental stress on DNA methylation in an invasive marine invertebrate, and found very rapid (<3 h) global DNA methylation differences in response to low salinity stress, but these differences had disappeared within 48 h (*Huang et al., 2017*). In this study, solitary ascidians (*Ciona savignyi*), were exposed to a lower salinity than the present study (20 PSU), with no gradual decrease allowing acclimation time, and all individuals died prior to the final sampling time point (120 h) (*Huang et al., 2017*). Studies of solitary ascidians have shown salinity exposure can induce a strong behavioural and physiological response in the first 48 h of severe osmotic stress, such as siphon closure and the excretion of intracellular osmolytes (*Toop & Wheatly, 1993*), which might be associated with such rapid, global DNA methylation changes. This may be a good strategy for surviving short-term non-optimal salinity events, such severe storms or transport through low salinity waters (*Rocha, Castellano & Freire, 2017*), but prolonged stress will likely lead to mortality. Nonetheless, this response demonstrates the potential for genome-wide methylation changes in response to salinity stress in an invasive marine ascidian.

*D. vexillum* is generally a cool water temperate species, but is found in a wide range of temperatures, from <0 - >24 °C (*Bullard et al., 2007*). Optimal temperature for growth appears to be between 14 °C and 20 °C. Based on this, we selected three temperature treatments: 19 °C (within the temperature range of *D. vexillum* colonies at the collection site, and the optimal range for *D. vexillum* globally), 25 °C (just outside the temperature range at the collection site, and near the upper limit globally), and 27 °C (+ 3 °C of

the upper limit at the collection site, but within global climate change predictions for the year 2100 (*IPCC, 2014*). Temperatures at the collection site in the Nelson marina typically range from 9 °C –10 °C (winter minima) and 22 °C –23 °C (summer maxima) (*Fletcher et al., 2013a*). The *D. vexillum* colonies used in this study would rarely experience temperatures of 25 °C or greater. The significant global DNA methylation changes observed in colonies held at 27 °C, after just three days exposure, is indicative of a dramatic response to thermal stress, a conclusion that is further supported by the significant negative growth of colonies held at these elevated temperatures compared to controls. Alterations to energetic balance (e.g., decreased growth/reproduction, switching to anaerobic metabolism) and protein expression profiles (e.g., upregulation of heat shock chaperones), are well known and energetically costly processes undertaken by invertebrates in response to thermal stress (*Sokolova et al., 2012*). Such responses can act to 'buy survival time' until conditions improve, and determine species distribution limits. This strategy may allow colonial ascidians to invade new areas. For example, the recent expansion of *D. vexillum* from temperate regions into the warmer, subtropical waters of the Mediterranean Sea (8–28 °C) provides evidence of the remarkable capacity of *D. vexillum* to adapt to increasing temperatures (*Ordóñez et al., 2015*). In temperate regions, maximum growth and reproduction in *D. vexillum* occurs during the warmer, summer months, with regression of colony growth and size occurring during winter (*Fletcher et al., 2013a*). In the Mediterranean, this cycle is reversed (*Ordóñez et al., 2015*). By growing and reproducing in the winter months and regressing during the summer months, *D. vexillum* is able to extend its introduced range towards warmer waters (*Ordóñez et al., 2015*).

The MSAP technique does not provide any insight into the identification of genes which are differentially methylated, so we are unable to demonstrate that any environmentally induced changes to DNA methylation are associated with functional traits that could lead to adaptive outcomes. However, correlative experiments have previously suggested a role for DNA methylation in adaptation to thermal stress, with natural populations of fish having higher levels of methylation in polar and sub-Antarctic species compared to temperate/tropical species (*Varriale & Bernardi, 2006*). Some natural populations of the solitary ascidian, *Ciona robusta*, display significant DNA methylation differences in genes that can be correlated with environmental differences in temperature and salinity (*Pu & Zhan, 2017*). Furthermore, experimental evidence for an adaptive response to temperature stress has been shown using an Antarctic marine polychaete worm, *Spiophanes tcherniai*. In this species, large DNA methylation shifts were observed after exposure to a 5.5 °C temperature increase, and this shift was accompanied by physiological adaptation, with respiration and metabolic rates returning to control levels in less than four weeks (*Marsh & Pasqualone, 2014*). Future studies utilising techniques with base pair resolution (e.g., bisulphite sequencing) will provide detailed insights into the location of methylation changes in specific genes associated with functional outcomes. This type of technique would also benefit from the analysis of downstream biological pathways, such as the analysis of gene expression and metabolomic profiling.

## CONCLUSIONS

In this study, we demonstrate the responsiveness of DNA methylation following exposure to an environmental gradient (temperature), which was correlated with phenotypic change (growth). Furthermore, DNA methylation changes did not occur in colonies exposed to an environmental gradient to which they may already be adapted (salinity). This is the first study to investigate DNA methylation patterns in a colonial ascidian, specifically the highly invasive *D. vexillum*, and adds to a growing body of evidence that DNA methylation plays a key role in the plasticity of adaptive traits. Epigenetic changes may contribute not only to the success of invasive species, but also to the adaptability of native species to changes within their environmental range. *D. vexillum* is an excellent model organism for future research into epigenetic responses to environmental stress. The responsiveness of DNA methylation to changes in the environment in this species lends itself to future studies testing the stability and longevity of these changes, and whether these changes can be associated with adaptive outcomes. Evidently, many key questions remain unanswered, including whether differences in methylation persist over time? Does tolerance for elevated temperature increase following exposure and, is tolerance associated with a specific epigenetic modification? However, our study establishes a baseline understanding of the role of DNA methylation in a globally invasive species. This unique study system provides a powerful framework for ecological epigenetic studies that could enhance our understanding of adaptation to rapid environmental change.

## ACKNOWLEDGEMENTS

We are grateful to Eric Goodwin (Cawthron) for help with R coding. Thank you to Achira Amadoru (Nelson Marlborough Institute of Technology) for assistance with experiment maintenance.

### Funding

This work was supported by the Marsden Fund of the Royal Society of New Zealand (CAW1401). The funders had no role in study design, data collection and analysis, decision to publish, or preparation of the manuscript.

### Grant Disclosures

The following grant information was disclosed by the authors:
Marsden Fund of the Royal Society of New Zealand: CAW1401.

### Competing Interests

Xavier Pochon and Andrew E. Fidler are Academic Editors for PeerJ.

### Author Contributions

- Nicola A. Hawes conceived and designed the experiments, performed the experiments, analyzed the data, prepared figures and/or tables, authored or reviewed drafts of the paper, approved the final draft.

- Louis A. Tremblay, Xavier Pochon, Brendon Dunphy and Andrew E. Fidler conceived and designed the experiments, authored or reviewed drafts of the paper, approved the final draft.
- Kirsty F. Smith conceived and designed the experiments, analyzed the data, contributed reagents/materials/analysis tools, authored or reviewed drafts of the paper, approved the final draft.

### Data Availability

The raw data are provided in the Supplemental File.

### Supplemental Information

Supplemental information for this article can be found online at http://dx.doi.org/10.7717/peerj.5003#supplemental-information.

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
