# Peer review of "Effects of temperature and salinity stress on DNA methylation in a highly invasive marine invertebrate, the colonial ascidian Didemnum vexillum"

_PeerJ, doi:10.7717/peerj.5003_

## Round 0.1 · original submission · Minor Revisions

Overall I enjoyed reading the manuscript of Hawes et al. The data presented highlighted that environmental temperature and salinity can influence DNA methylation in an invasive species. The study is very well written and set out. There a few minor areas the paper could be improved upon and these include:

1) More justification of the choice of ascidians as models for invasive species
2) The methods and results could be a little clearer in places as pointed out by reviewer 2
3) Line 220 should be Tukey not Tuckey

Please respond to all of the reviewers comments in your revision.

Reviewer 1 ·

Basic reporting

The authors investigate DNA methylation in rapid response to environmental stress. The manuscript is quite well written. It is clear and unambiguous, professional English is used throughout. The article conforms to professional standards. It cites relevant literature and sufficient background is provided for readers from diverse backgrounds. The work is also strongly connected to a broad filed of knowledge and a widely-interesting research area.

The manuscript follows professional article structure, sufficient information is shared to judge the conclusions. All tables and figures are easy to follow and present necessary information. It is self-contained and includes all relevant results necessary to support the hypothesis.

Experimental design

The manuscript describes original research that fits the aimss of the journal. The research question is well defined and fills an identified knowledge gap. The research is highly rigorous to a high technical & ethical standard. All techniques are well described and used accurately, following common standards. The methods well described, and no issues exist with what was done.

Validity of the findings

The results are valid and meaningful to a wide audience. Data are robust, methods and analysis were statistically sound, the experiment followed proper controls.
The conclusions were clear and well supported. There is little issue with the manuscript in general and the results and conclusions are quite strong.

Additional comments

I found this manuscript to be extremely well written. The methods, results, and conclusions were sound. I found no issues with the presentation. I think the results are quite interesting. The use of MS-AFLP was appropriate and generated an interesting finding that supports a growing field of research.

·

Basic reporting

The manuscript presents an interesting study documenting rapid DNA methylation response to the effects of temperature and salinity stresses in the highly invasive ascidian Didemnum vexillum. This manuscript is well written and easy to follow. However, some modifications on tables and figures are still needed. In Table 1, the ‘/’ in selective primers HapII + TCC is unnecessary. In Figure 4, ‘Time 0’, ‘Time 1’, ‘Time 2’ should be labelled on A), B), C), respectively, to increase the readability. In supplementary Table S9, there may have some errors in the details of sampling times, as ‘T3’ was not referred in this paper. In addition, lines 86-90, I suggest to add several sentences to summarize why ascidians are good models for studying invasion success..

Experimental design

This manuscript takes advantage of asexual reproduction to reduce the possible effects of genetic variation on epigenetic results. However, I got confused why authors conducted the analysis of molecular variance (AMOVA) for NML at different treatments and different sampling time points (TableS4, TableS6, TableS9, TableS11). Colonies allocated in each treatment are assumed to be derived from the same source and the genetic background should be very similar. Furthermore, all colonies survived under stresses, which may suggest both temperature and salinity do not have a selective effect on genetic variation. So the results of AMOVA among groups and within each group should be not significant. However, significant different results were observed between differential salinity treatments at T0 in NML (lines 271-272). This may be due to the very different genetic background of samples (lines 128-129 it may be two or more colonies growing in close proximity). So, it is needed to further discuss how to control the genetic effects on epigenetic results. In addition, based on statistical methods, NML fragments were also produced by methylated sensitive enzymes but classified as non-methylated loci. Therefore, NML may also include some epigenetic information when they represent genetic variation, and the significantly different results of NML may be the reflection of epigenetic variation in another form.

Validity of the findings

In this paper, the number of MSL was dynamic at different time points. For example, in the temperature stress experiments, 586 of the 1157 loci were MSL at T0 while 613 loci were MSL at T1 (lines 235-240). In the R package msap, loci are classified into MSL and NML based on a threshold (the default is 0.05). Therefore, different datasets would result in different numbers of MSL and NML statistically. However, loci, which represent genetic variation or epigenetic variation, are their own in inherent characteristics. It doesn’t make sense that the same loci represent genetic variation (NML) at T0 but represent epigenetic variation (MSL) at T1. I recommend the mixing scoring approach (Schulz et al., 2013) to analyze MSAP data, which may be more suitable for this study.

Additional comments

In the study of Pu & Zhan 2017, several methylation differences have been associated with environmental differences in salinity and temperature (lines 312-315), but it cannot be stated that the temperature- and salinity-related methylation modifications are very small in the genome as it is a locus-specific but not genome-wide level study.

---

## Round 0.2 · accepted · Accept

In my opinion this well written and thought out manuscript should be accepted as is. The current version of the manuscript has addressed all the concerns raised by the two reviewers and myself. This is a substantial piece of work that I feel will make a solid contribution to not only PeerJ, but the invasion genomics field in general.

# ·

Basic reporting

The authors have addressed all my concerns that I raised during the first round of review. Thus, I suggest "acceptance" for publication in PeerJ.

Experimental design

no comment

Validity of the findings

no comment

Additional comments

no comment